# Exploring the Anti-Colorectal Cancer Mechanism of Norcantharidin Through TRAF5/NF-κB Pathway Regulation and Folate-Targeted Liposomal Delivery

**DOI:** 10.3390/ijms26041450

**Published:** 2025-02-09

**Authors:** Fanqin Zhang, Xiaodong Chen, Chuanqi Qiao, Siyun Yang, Yiyan Zhai, Jingyuan Zhang, Keyan Chai, Haojia Wang, Jiying Zhou, Meiling Guo, Peiying Lu, Jiarui Wu

**Affiliations:** Department of Clinical Chinese Pharmacy, School of Chinese Materia Medica, Beijing University of Chinese Medicine, Beijing 102488, China; 20210941446@bucm.edu.cn (F.Z.); 20240941540@bucm.edu.cn (X.C.); 20230935191@bucm.edu.cn (C.Q.); 20230935192@bucm.edu.cn (S.Y.); 20210935168@bucm.edu.cn (Y.Z.); 20210941447@bucm.edu.cn (J.Z.); 20220935196@bucm.edu.cn (K.C.); 20230941519@bucm.edu.cn (H.W.); 20230935269@bucm.edu.cn (J.Z.); 20240935280@bucm.edu.cn (M.G.); 20240935210@bucm.edu.cn (P.L.)

**Keywords:** colorectal cancer, norcantharidin, TRAF5/NF-κB pathway, liposomes

## Abstract

Colorectal cancer is one of the most common malignant tumors worldwide, significantly impacting human health. Cantharidin (CTD), an active compound derived from the Spanish fly, exhibits antitumor properties. Its derivative, norcantharidin (NCTD), is synthesized by removing methyl groups from positions 1 and 2 of cantharidin. NCTD has demonstrated lower toxicity while maintaining similar antitumor effects compared to CTD. However, the mechanism by which NCTD exerts its effects against colorectal cancer remains unclear. Here, we conducted a comprehensive analysis of the effects of NCTD on colorectal cancer both in vitro and in vivo. Whole-transcriptome sequencing and bioinformatics tools were employed to identify potential key targets of NCTD in the treatment of colorectal cancer. Additionally, we designed folate-receptor-targeting NCTD liposomes (FA-NCTD) and assessed their anticancer efficacy in vivo. NCTD effectively inhibited cell viability, clonal formation, and migration in HCT116 and HT-29 cell lines. NCTD also induced apoptosis, influenced the cell cycle, altered mitochondrial membrane potential, and increased reactive oxygen species levels. The whole-transcriptome sequencing and bioinformatics analysis identified TRAF5 as a key target for NCTD’s action against colorectal cancer. Furthermore, NCTD was found to regulate the TRAF5/NF-κB signaling pathway in both HCT116 and HT-29 cells. The FA-NCTD liposomes demonstrated effective tumor targeting and significantly inhibited tumor growth in vivo. This result showed that NCTD effectively suppresses the malignant proliferation of colon cancer cells by modulating the TRAF5/NF-κB signaling pathway and inducing programmed apoptosis, thereby offering a novel strategy for colorectal cancer treatment. The prepared FA-NCTD liposomes provide a promising approach for achieving the precise targeting and controlled release of NCTD.

## 1. Introduction

Colorectal cancer (CRC) is one of the common malignant tumors of the digestive system [1]. Colorectal cancer poses a critical hazard to patients’ lives and healthcare due to its high incidence and mortality [2]. The disease originates from the abnormal proliferation of the colonic mucosa epithelium, and its course often goes through a continuous process from benign polyps and adenomatous changes to high-grade dysplasia and, finally, to cancer [3]. The pathogenesis of colon cancer is a complex, multi-factor, and multi-step process, involving the interaction of genetics, environment, diet, and lifestyle [4]. The early symptoms of colon cancer are not obvious and are often ignored. Drug resistance and the obvious side effects of radiotherapy and chemotherapy cause a large number of colon cancer patients to lose their lives every year [5]. Therefore, it is vital to accelerate the research and development of novel diagnostic markers, early screening methods, and innovative treatment strategies for colon cancer to improve the survival rate and quality of life.

Traditional Chinese medicine has developed into one of the key clinical antitumor therapeutic modalities [6]. Cantharidin, in the form of the dried bodies of the Chinese blister beetles Mylabris phalerata and M. cichorii, displays antitumor activity and induces apoptosis in many types of tumor cells [7]. Norcantharidin (NCTD) has a lower toxicity and similar antitumor effects in comparison with cantharidin. NCTD plays a therapeutic role in various tumors, such as hepatocellular carcinoma [8], neuroblastoma [9], and bladder cancer [10]. In the treatment of colorectal cancer, norcantharidin, as an effective antitumor agent, induces colorectal cancer cell apoptosis [11,12]. Many pathways are confirmed to be engaged in NCTD’s effects against colorectal cancer, among which are cell apoptosis, anti-proliferative pathways, anti-angiogenesis pathways, and epithelial–mesenchymal transition [11,13,14,15]. However, the mechanism of NCTD in colorectal treatment remains unclear. In addition, in order to reduce NCTD’s toxicity and improve its efficacy, designing targeted drug delivery systems is one of the most feasible strategies. Liposome technology, as an advanced drug carrier system with a unique phospholipid bilayer structure, separately encapsulates hydrophilic and hydrophobic drugs [16]. Due to its similarity with biofilm components, liposome technology shows good biocompatibility and circulation stability [17]. Due to the overexpression of folate receptors on the surface of tumor cells, folate receptors have become important targets for anticancer therapy [18].

This study systematically revealed the potential efficacy and pharmacological mechanism of NCTD in the treatment of colon cancer through in vivo and in vitro experiments, high-throughput sequencing technology, and molecular biology experiments. The antitumor effects of NCTD were confirmed in vivo and in vitro. Whole-transcriptome sequencing showed that TRAF5 was closely related to NCTD in colorectal cancer. NCTD was further shown to regulate the TRAF5/NF-κB pathway to inhibit the malignant progression of colon cancer. In addition, NCTD liposomes targeting the folate ligand (FA-NCTD) were designed and synthesized, and the targeting and anti-colon cancer effects of the liposomes in vivo were further explored. These results provide a new strategy and theoretical basis for the treatment of colon cancer.

## 2. Results

### 2.1. NCTD Inhibited Cell Proliferation and Migration in CRC Cells

The effect of NCTD on cell viability was investigated in HCT116 and HT-29 cells treated with different concentrations of NCTD. The structure of norcantharidin is shown in Figure 1A. After being incubated for 24 h, 48 h, and 72 h, NCTD reduced cell viability in both a dose- and time-dependent manner (Figure 1B). The NCTD IC_50_ values for HCT116 were 104.27 ± 13.31, 54.71 ± 4.53, and 37.68 ± 3.92 μM at 24 h, 48 h, and 72 h, respectively. The NCTD IC_50_ values for HT-29 were 118.40 ± 6.06, 41.73 ± 7.69, and 24.12 ± 1.37 μM at 24 h, 48 h, and 72 h, respectively. As shown in Figure 1C, the colony formation assay results showed that the number of former colonies significantly decreased with increasing NCTD concentrations in HCT116 and HT-29 cells, further indicating the anti-proliferative effect of NCTD. In addition, the morphology of HCT116 and HT-29 cells changed with the increase in NCTD concentrations, which was mainly characterized by cell contraction and rounding, larger cell spacing, smaller cell density, and more floating cells (Figure 1D). With the increase in concentration and time, two cell-line morphological changes became more obvious (Figure 1C). The wound-healing assays revealed that NCTD dose-dependently suppressed HCT116 and HT-29 cell migration at 24 h and 48 h (Figure 1E). Taken together, these results indicate that NCTD inhibited cell proliferation and migration in CRC cells.

### 2.2. NCTD Induced Apoptosis and Cell Cycle Arrest

To explore the effect of NCTD on apoptosis and cell cycle arrest, the staining of the cells was performed with Annexin V/Protium Iodide. The results showed that NCTD significantly promoted the apoptosis of HCT116 and HT-29 cells after 48 h (Figure 2A). The apoptosis rates were up to 44.23 ± 1.11% and 46.43 ± 5.22% in HCT116 and HT-29 cells treated with 120 μM NCTD. Moreover, G0-G1 arrest in HCT116 cells was stimulated after treatment with NCTD for 48 h. Compared with the control group, the proportion of G2-M phase cells increased significantly when treated with 50 μM NCTD (Figure 2B). The proportion of S phase cells decreased when treated with 30 μM and 50 μM NCTD. In HT-29 cells, the proportion of cells in the G0-G1 phase decreased, and those in the G2-M phase increased after treatment with 40 μM and 60 μM NCTD (Figure 2B). Overall, NCTD induced cell cycle arrest and apoptosis in HCT116 and HT-29 cells.

### 2.3. NCTD Induced Mitochondrial Dysfunction and Enhanced Reactive Oxygen Species (ROS) Levels

Changes in mitochondrial membrane potential may be triggered when cells undergo apoptosis. To investigate the effect of NCTD on mitochondrial membrane potential, the ratio of green and red fluorescence intensity was measured using a JC-1 fluorescent probe (Beyotime, Shanghai, China). The results indicated that the ratio of green and red fluorescence increased after treatment with different concentrations of NCTD in HCT116 and HT-29 cells (Figure 2C). NCTD induced the depolarization of mitochondrial membrane potential, suggesting that it may induce mitochondrial apoptosis in colon cancer cells. When cells are under oxidative stress, large amounts of ROS are produced within the mitochondria, resulting in a decrease in mitochondrial membrane potential. In turn, it further increases the production of ROS in a vicious cycle. The mitochondrial membrane potential is closely associated with ROS levels. As shown in Figure 2D, the fluorescence intensity of ROS in low-, medium-, and high-dose NCTD was significantly increased compared with the control group. These findings suggest that NCTD mediated mitochondrial dysfunction by inducing intracellular ROS production, thus promoting the apoptosis of colon cancer cells.

### 2.4. Bioinformatics Analysis of NCTD Intervention HCT116 and HT-29 Cells

To further explore the molecular mechanisms of NCTD’s anti-colon cancer effects, mRNA-seq data and clinical information on colon cancer from The Cancer Genome Atlas (TCGA) database were downloaded. This was carried out to investigate changes in gene expression and gain a comprehensive understanding of the potential pathogenesis and prognostic information of colon cancer. Compared to adjacent non-cancerous samples, 6220 genes were upregulated, and 5622 genes were downregulated in tumor tissues (Figure 3A). GO and KEGG enrichment analysis were performed for up-regulated and down-regulated genes in tumor samples (Appendix A). Subsequently, the Cox proportional hazards model was employed to assess the impact of each gene on the prognosis of cancer patients. Among them, 1750 genes with HR > 1 and *p* < 0.05 were considered risk factors for cancer patients, prolonging the overall survival of patients, whereas 205 genes with HR < 1 and *p* < 0.05 were deemed protective factors, favorable for extending the overall survival of patients. To explore the regulatory effects of norcantharidin on gene expression and its potential functional impacts more deeply, we conducted Gene Set Enrichment Analysis (GSEA) of the differentially expressed genes in HCT116 and HT-29 cell lines treated with norcantharidin. During the screening process, we adopted rigorous statistical criteria, namely, an adjusted *p*-value (*p*.adjust) of less than 0.05 and a q-value of less than 0.25, to ensure the statistical significance of the identified enriched gene sets. The analysis revealed that the gene sets significantly enriched in both cell lines are closely associated with tumorigenesis and development, further highlighting the potential importance of norcantharidin in tumor biology. Appendix A presents the top five gene sets ranked by their Normalized Enrichment Score (NES). A forest plot was used to display the top 10 genes with the smallest *p*-values (Figure 3B). Whole-transcriptome sequencing was conducted on HCT116 and HT-29 cells. Comparing the HCT116 cells treated with norcantharidin to untreated cells, a total of 7831 differential genes were identified, with 3952 upregulated and 3879 downregulated after treatment (Figure 3C). Similarly, in HT-29 cells treated with norcantharidin versus untreated cells, 6205 differential genes were found, including 3096 upregulated and 3109 downregulated genes after treatment (Figure 3D). GO and KEGG enrichment analysis were performed for up-regulated and down-regulated genes in HCT116 cells and HT-29 (Appendix A).By intersecting the pathologically differential genes, pharmacologically differential genes, and genes with survival significance, a total of 40 key genes were obtained (Figure 3E). The protein–protein interaction analysis and Kyoto Encyclopedia of Genes and Genomes (KEGG) enrichment analysis of these key genes revealed that TRAF5 is closely related to the occurrence and development of cancer and is involved in the activation of the NF-κB signaling pathway (Figure 3F,G).

### 2.5. NCTD Inhibits Colon Cancer by Regulating the TRAF5/NF-κB Signaling Pathway

According to the above results, TRAF5 plays a key role in NCTD intervention in colorectal cancer. TRAF5 is involved in the activation of the NF-κB signaling pathway. Therefore, the effect of NCTD on the TRAF5/NF-κB signaling pathway was investigated in HCT116 and HT-29 cells. After treatment with NCTD, the mRNA levels of TRAF5, IκBα, and p65 decreased in two cell lines (Figure 4A). As shown in Figure 4B, NCTD exhibited the protein expression of TRAF5, *p*-IκBα, and *p*-p65 in HCT116 and HT-29 cells. However, the protein expression of IκBα and p65 remained basically invariant.

The mRNA and protein expressions of c-Myc involved in the regulation of many genes related to cell proliferation were reduced in NCTD-treated HCT116 and HT-29 cells (Figure 4C,D). Meanwhile, NCTD significantly inhibited the expression of CDK4, CDK6, and Cyclin D1 protein in HCT116 (Figure 4D). The mRNA and protein expression of CDK1 and Cyclin B1 were dose-dependently restrained in HT-29 cells (Figure 4D). In addition, NCTD increased the mRNA and protein levels of Bax in HCT116 and HT-29 cells, while the expression of Bcl-2 was observed (Figure 4E,F). The mRNA and protein expressions of Cyt-C and Caspase-9, -3, and -8 increased in NCTD-treated HCT116 and HT-29 cells (Figure 4E,F). The levels of XIAP and cFLIP were inhibited (Figure 4E,F). In NCTD-treated CHT116 and HT-29 cells, the mRNA and protein of cleaved caspase-9, -3, and -8 significantly increased (Figure 4F).

### 2.6. NCTD Suppressed the Progression of Colon Cancer Transplanted in Nude Mice

To determine the dosage of NCTD in vivo, the effect of toxicity was examined in nude mice treated with saline and NCTD (5, 10, 20, and 40 mg/kg) (Figure 5A). In the experiment, all nude mice in the NCTD-20 mg/kg and NCTD-40 mg/kg groups died on the first day of administration. The survival of the mice administered saline, NCTD-5 mg/kg, and NCTD-10 mg/kg was good. Therefore, the maximum dose of NCTD was found to be 10 mg/kg in vivo. The administration time, dosage, and method are shown in Figure 5B. Oxaliplatin and NCTD-10 mg/kg showed inhibitory effects on tumor cells (Figure 5C,D). On day 13, oxaliplatin and NCTD-10 mg/kg exhibited similar inhibitory effects (Figure 5E). Tunnel-positive cells and cleaved caspase-3-positive cells increased in the oxaliplatin and NCTD-10 mg/kg groups (Figure 5F). However, TRAF5 and the *p*-p65-positive ratio were reduced (Figure 5F). Compared with the saline group, the number of Ki67-positive cells declined in the oxaliplatin, NCTD-5 mg/kg, and NCTD-10 mg/kg groups (Figure 5F). There were no significant changes in terms of body weight or organ coefficient in the heart, liver, spleen, lung, and kidney (Figure 5G–I).

### 2.7. Preparation and Quality Evaluation of NCTD Liposomes Targeting Folate Receptors

In view of their overexpression on the surface of tumor cells, folate receptors have become an important target for anticancer therapy. Liposomes loaded with NCTD targeting folate ligands were designed and synthesized that specifically recognize and bind to folate receptors on the surface of tumor cells, so as to achieve the precise targeted aggregation and release of drugs. The maximum absorption wavelength of the NCTD solution was 190 nm, as measured using a UV2600 ultraviolet spectrophotometer (Figure 6A). The absorbance of NCTD is linear in the concentrations of 10–160 μM (Figure 6A). The encapsulation rate of NCTD liposomes is shown in Appendix A. When the mass ratio of cholesterol, NCTD, and DSPE-PEG_2000_ was 5:1:20, and following ultrasonic for 30 min, the optimal encapsulation rate was 43.6%. The ratio of cholesterol, NCTD, DSPE-PEG_2000_, and DSPE-PEG_2000_-FA was 5:1:10:10, and via ultrasonic treatment for 30 min, FA-NCTD liposomes with optimum encapsulation rates were prepared for subsequent experiments (Appendix A). Transmission electron microscopy (TEM) showed that the NCTD, FA-NCTD, IR-1061-NCTD, and IR-1061-FA-NCTD liposomes were round in shape and uniform in size (Figure 6B). The particle size and Zeta potential of the liposomes were measured using a dynamic light scatterer. As shown in Figure 6C, the particle size of the NCTD, FA-NCTD, IR-1061-NCTD, and IR-1061-FA-NCTD liposomes was 95.4 ± 1.9 nm, 105.1 ± 4.2 nm, 122.4 ± 3.8 nm, and 129.1 ± 1.4 nm, respectively. The Zeta potentials of the four liposomes were −11.5 ± 2.4 mV, −10.1 ± 3.4 mV, −6.7 ± 1.1 mV, and −5.4 ± 1.4 mV, respectively.

There was no significant change in the absorbance values before and after the centrifugation of the NCTD and FA-NCTD liposomes (Figure 6D). The diameter of the NCTD and FA-NCTD liposomes did not change significantly at 4 °C and 37 °C (Figure 6E). With the extension of the observation time at 4 °C and 37 °C, the diameter of the liposomes remained steady (Figure 6E). IR-1061-FA-NCTD liposomes showed strong fluorescence in the tumor at 4 h, 8 h, 12 h, and 24 h compared with IR-1061-NCTD liposomes (Figure 6F). These results indicate that folate-functionalized liposomes exhibit superior tumor-targeting capability. These results indicate that the IR-1061-FA-NCTD liposomes rely on folic acid modification on the surface of the liposomes to become more enriched at the tumor site, proving that the use of targeted liposomes is feasible and that the prepared liposomes can increase the drug concentration at the tumor site to achieve better therapeutic effects.

### 2.8. FA-NCTD Liposomes Suppressed the Growth of Colon Cancer Transplanted in Nude Mice

To further study the anti-colon cancer effect of FA-NCTD liposomes in vivo, the inhibitory effect of liposomes on transplanted tumors in mice was investigated. The method, dosage, and time of administration are shown in Figure 7A. As shown in Figure 7B–D, FA-NCTD liposomes have the strongest inhibitory effect on tumor growth. The immunofluorescence results showed that the Ki67-, TRAF5-, and *p*-p65-positive ratios were reduced in the oxaliplatin, NCTD, NCTD liposome, and FA-NCTD liposome groups (Figure 7E). FA-NCTD liposomes had a better inhibitory effect on tumor growth. In addition, the mice in the saline and oxaliplatin groups exhibited slight weight loss (Figure 7F), although the other groups demonstrated no significant change in body weight. The histological results indicated that there were no significant pathological changes in the heart, liver, spleen, lungs, or kidneys of the mice (Figure 7G), and no significant changes in organ coefficients in these organs (Figure 7H).

## 3. Discussion

Colon cancer is a common malignant tumor with high incidence and mortality rates worldwide [19]. The occurrence of colon cancer is mainly related to diet, lifestyle, environment, genetic factors, and intestinal microbial disorders [20]. The most common chemotherapy regimens for colorectal cancer are oxaliplatin, fluorouracil, and irinotecan complemented by radiotherapy, immunotherapy, and molecular-targeted strategies [21,22]. However, drug resistance and side effects are the main problems in clinical application. The rapid progress in modern pharmacology and molecular biology has unveiled the remarkable antitumor properties of chemical compounds in numerous traditional Chinese medicines (TCMs), offering a wealth of resources for anticancer drug discovery. Among them, cantharidin, the primary bioactive component extracted from Mylabris, exhibits potent inhibitory activity against various tumor cell lines [23,24,25,26]. However, its clinical application is hampered by significant toxic and side effects in vivo [27,28,29]. Norcantharidin (NCTD), a structurally modified derivative of cantharidin with the methyl groups at the 1,2 positions removed, demonstrates reduced toxicity while retaining similar antitumor efficacy. Many studies have revealed that NCTD exerts its antineoplastic effects by modulating diverse signaling pathways and inhibiting the progression of multiple malignancies [30,31,32]. Despite its promising potential as a novel anticancer agent, research specifically investigating the role of NCTD in colorectal cancer treatment remains insufficient, warranting further exploration. The finding of this study was the inhibitory effects of NCTD on the growth and proliferation of HCT116 and HT-29 cells (Figure 1). Norcantharidin induced cell apoptosis and cycle arrest in HCT116 and HT-29 cells (Figure 2A,B). Early apoptotic events are characterized by cellular shrinkage, a rounded morphology, loss of surface microvilli, and progressive detachment from neighboring cells, consistent with the observations in our study [33]. The apoptosis rates in colorectal cancer cells were significantly elevated after NCTD treatment, indicating that apoptosis induction may be the antitumor mechanism. The process of apoptosis primarily proceeds via intrinsic and extrinsic pathways [34]. Within the intrinsic pathway, mitochondria act as the central regulator, with alterations in mitochondrial membrane potential serving as a key event in initiating apoptosis [35]. The depolarization of mitochondrial membrane potential and the increase in ROS levels induced by NCTD contributed to its antitumor effect (Figure 2C,D). Bcl-2 and Bax, two key members of the Bcl-2 protein family, regulate the opening and closing of MPTPs. Pro-apoptotic proteins such as Bax promote MPTP opening by interacting with the adenine nucleotide translocator (ANT) on the mitochondrial inner membrane and the voltage-dependent anion channel (VDAC) on the outer membrane. In contrast, the anti-apoptotic protein Bcl-2 competitively binds to ANT or VDAC, inhibiting MPTP opening [36]. Once the MPTP is open, Cyt-C is released into the cytosol, activating the Cyt-C/Caspase-9 signaling pathway, which leads to mitochondrion-dependent apoptosis. Dysregulated c-Myc expression drives tumor cells to escape normal proliferative constraints, promoting uncontrolled growth [37,38,39]. NCTD significantly downregulates the expression of c-Myc in colon cancer cell lines (Figure 4). In the normal cell cycle, upon receiving mitotic signals, Cyclin D is upregulated and forms a complex with CDK4/6, known as the Cyclin D-CDK4/6 complex. This complex phosphorylates the Retinoblastoma protein (RB), leading to the release of the transcription factor E2F, which is a crucial step in the transition from G1 to S phase. In the G2 phase, the synthesis of Cyclin B1 reaches its peak and binds to CDK1 to form the Cyclin B1-CDK1 complex, also referred to as the maturation-promoting factor (MPF). Once this complex reaches a certain threshold, it triggers the cell’s entry into the M phase. The activation of Cyclin B1-CDK1 is a key regulatory step in the G2/M phase transition [40]. In this study, the results showed that NCTD significantly reduced the expression levels of Cyclin D1, CDK4, and CDK6 in HCT116 cells, thereby inducing G2/M phase arrest. Conversely, in HT-29 cells, NCTD downregulated Cyclin B1 and CDK1, resulting in G0/G1 phase arrest. These findings highlight the ability of NCTD to induce cell cycle arrest at distinct phases in different cell lines, possibly reflecting species- or cell-line-specific biological variations. X-linked inhibitor of apoptosis protein (XIAP), a potent member of the inhibitor of apoptosis protein (IAP) family, directly inhibits caspases and regulates apoptosis through multiple mechanisms [41,42]. Treatment with NCTD decreased the expression of XIAP in HCT116 and HT-29 cells, further confirming that NCTD induces apoptosis in colorectal cancer cells.

The TNF receptor-associated factor (TRAF) family consists of intracellular adaptor proteins characterized by a conserved C-terminal domain. TRAF5 is closely related to the occurrence and development of cancer. TRAF5 is closely associated with the TNF receptor superfamily and plays a pivotal role in cellular signal transduction, particularly in regulating processes such as cell proliferation, differentiation, and apoptosis. Studies demonstrated that TRAF5 is closely linked to the onset and progression of cancer [43]. Wang et al. [44] found that the overexpression of TBC1D3 in breast cancer cells activated the NF-κB signaling pathway and upregulated TRAF5, promoting cell migration. Wu et al. [45] reported higher TRAF5 expression in hepatocellular carcinoma (HCC), such as HepG2, HuH7, SMMC-LM3, and Hep3B. Zhang et al. [46] found that the long non-coding RNA (lncRNA) HCG18 promoted epithelial ovarian cancer cell proliferation, migration, and epithelial–mesenchymal transition (EMT) by targeting miR-29a/b and upregulating TRAF4/TRAF5. TRAF5 is a direct target of miR-873 in colorectal cancer, inhibiting cancer cell proliferation [47]. NF-κB, a protein complex, primarily regulates the transcription of DNA and controls the production of cytokines, as well as influencing cell survival [48]. In tumor progression, the NF-κB signaling pathway is often persistently activated, leading to the abnormal expression of NF-κB [49]. This promotes tumor cell proliferation, inhibits apoptosis, and facilitates angiogenesis and metastasis [50,51,52]. TRAF5 can activate the NF-κB pathway by interacting with the IKK (IκB kinase) complex, promoting IKK activation. This activation leads to the phosphorylation and degradation of IκB (NF-κB inhibitor), allowing NF-κB dimers to enter the nucleus, where they bind to specific DNA sequences and activate the transcription of genes associated with cell growth [53,54,55,56]. We found that NCTD downregulates TRAF5 expression, subsequently inhibiting the phosphorylation of IκBα and p65 in colorectal cancer cells, thereby affecting their growth and survival. Furthermore, NCTD inhibited tumor progression without causing significant organ toxicity in vivo.

These drugs leverage the molecular differences between tumor cells and normal cells, allowing for specific binding to and targeting of tumor cells for precise treatment. This strategy has inspired the development of liposomal targeted delivery systems, in which targeting molecules (such as antibodies or ligands) are attached to the liposome surface, enabling the system to actively recognize and bind to tumor cells for efficient drug delivery. Due to the overexpression of folate receptors on the surfaces of tumor cells, these receptors have emerged as significant targets for cancer therapy. In this study, we employed PEG modification technology combined with a folate-targeted strategy to design and synthesize actively targeted liposomes loaded with NCTD. The folate ligand on the liposome surface specifically targets and binds to folate receptors on colon cancer cells, ensuring precise drug delivery and release. This approach aims to enhance the antitumor efficacy of NCTD while minimizing its potential toxic effects on non-target tissues. This strategy aims to develop targeted liposomes that incorporate tumor-cell-specific ligands, enhancing the accumulation of drugs at the tumor site for more effective anticancer treatment. Furthermore, targeted liposomes containing NCTD may exhibit synergistic effects when used in combination with other anticancer drugs, thereby enhancing overall efficacy. Norcantharidin has fewer side effects than drugs such as oxaliplatin and fluorouracil, potentially improving patient tolerance to the drug. However, more extensive clinical studies are still needed to verify its efficacy and safety in the treatment of colorectal cancer.

## 4. Materials and Methods

### 4.1. Reagents

Norcantharidin (NCTD) was purchased from Shanghai Yuanye Bio-Technology Co., Ltd. (Shanghai, China). Cell Counting Kit-8 (CCK-8) was obtained from Biorigin Co., Ltd. (Beijing, China). Crystal violet dye and 4% Tissue Fix Solution were purchased from Beijing Solarbio Science & Technology Co., Ltd. (Beijing, China) An Annexin V-FITC Apoptosis Assay Kit was provided by Becton, Dickinson and Company (Franklin Lakes, NJ, USA). A Cell Cycle Analysis Kit, Mitochondrial Membrane Potential Kit, and Reactive Oxygen Species Assay Kit were supplied by Beyotime Biotechnology Co., Ltd. (Shanghai, China).

### 4.2. Cell Lines and Culture

Human colon cancer cell lines (HCT116 and HT-29) were purchased from Pricella Biotechnology Co., Ltd. (Wuhan, China). Both cells were cultured in McCoy’s 5A culture medium containing 10% fetal bovine serum (FBS) and 1% penicillin–streptomycin (PS) at 37 °C in an atmosphere of 5% CO_2_.

### 4.3. Cell Viability

A Cell Counting Kit-8 was used to monitor cell viability. Cells were planted in 96-well plates at a density of 1 × 10^4^ cells/mL and attached for 24 h. After incubation with different concentrations of norcantharidin, 10 μL of CCK-8 solution was added to each well at 24, 48, and 72 h. After 1 h, the absorbance value was measured at 450 nm using a microplate reader (SpectraMax i3x, San Jose, CA, USA).

### 4.4. Colony Formation

Cells were seeded into 6-well plates for cell adherence. Cells were cultured in various concentrations of norcantharidin for 10 to 14 days. After the termination of the culture, 4% Tissue Fix Solution was added. Then, the cells were stained with 0.1% crystal violet solution.

### 4.5. Cell Morphology

For observing cell morphology, HCT116 and HT-29 cells were seeded in 6-well plates. After adhesion, the cells were treated with various concentrations of norcantharidin. The morphological changes and growth of the cells were observed using an inverted microscope (Nikon, Tokyo, Japan).

### 4.6. Wound-Healing Assay

HCT116 and HT-29 cells were planted in a 6-well plate at a density of 5 × 10^5^ cells/mL. When the cells were fused to 90–100%, a wound was created with 200 μL sterile pipette tips. The effects of different concentrations of norcantharidin on the wound were observed under an inverted microscope after 24 h and 48 h. Image J was used to measure the scratch area.

### 4.7. Cell Apoptosis Assay

HCT116 and HT-29 cells were seeded and incubated with various concentrations of norcantharidin for 48 h. Then, 5 μL of PI was added after cells were stained with 5 μL of Annexin V-FITC. Cells were instantly detected via flow cytometry.

### 4.8. Mitochondrial Membrane Potential

HCT116 and HT-29 cells were collected after being treated with norcantharidin for 48 h. The cell precipitates were re-suspended in 500 μL medium, and then 500 μL JC-1 staining solution was added. The mixture was incubated at 37 °C for 20 min. Then, the cells were analyzed using flow cytometry. The changes in mitochondrial membrane potential were assessed by calculating the JC-1 green/red fluorescence ratio.

### 4.9. Cell Cycle Analysis

The effect of norcantharidin on the cell cycle of HCT116 and HT-29 cells was detected using a PI staining assay. After the cells were exposed to norcantharidin for 48 h, the cells were washed with PBS and fixed with 70% anhydrous ethanol for 24 h. Then, the cells were stained with a dye solution containing PI and kept in the dark for 30 min.

### 4.10. Reactive Oxygen Species (ROS)

HCT116 and HT-29 cells were incubated with norcantharidin for 48 h. Both cell types were harvested and incubated with DCFH-DA for 20 min. After incubation, the cells were washed three times with a serum-free medium to remove DCFH-DA that had not penetrated the cell interior. Then, fluorescence intensity was measured using flow cytometry (CytoFLEX, Brea, CA, USA).

### 4.11. Whole-Transcriptome Sequencing

Firstly, HCT116 and HT-29 cells in the logarithmic growth phase were inoculated into T75 flasks. Then, the cells were treated with 60 μM and 40 μM of norcantharidin, respectively, for a duration of 48 h. Afterward, the cells were lysed with TRIzol reagent and collected into RNase-free tubes for storage. Subsequently, RNA was extracted from the cells and underwent strict quality control, including agarose gel electrophoresis, a purity assessment using a NanoPhotometer spectrophotometer, and integrity evaluation with an Agilent 2100 bioanalyzer (Santa Clara, CA, USA). Next, mRNA was purified using Oligo(dT) magnetic beads, followed by fragmentation. Double-stranded cDNA was synthesized from the fragmented mRNA using random primers through reverse transcription. The cDNA underwent end-repair and adapter ligation. AMPure XP beads were used to select cDNA fragments of appropriate size, and a cDNA library was constructed through PCR enrichment. After library quality assessment—including initial quantification with a Qubit2.0 Fluorometer (Thermo Fisher Scientific, Waltham, MA, USA), insert size detection with an Agilent 2100 bioanalyzer, and the accurate quantification of effective concentrations using RT-qPCR (with the effective concentration needing to be higher than 2 nM)—qualified libraries were pooled based on their effective concentration and target data output for Illumina sequencing. The sequencing principle is sequencing by synthesis, where sequence information is obtained by capturing fluorescent signals.

### 4.12. Bioinformatics Analysis

Firstly, RNA-seq expression profiles of 499 colon cancer patients were downloaded from the TCGA database, and 41 paired samples were extracted from the expression matrix. Differential analysis was then performed on these samples using the R package DESeq2 (Version 1.32.0), with a fold change ≥1.2 and *p* < 0.05 as the screening criteria, to identify differentially expressed genes between paired tumor and adjacent non-tumor samples. Subsequently, the same method was utilized to analyze the differentially expressed genes from whole-transcriptome sequencing. Next, the R package survival was employed to integrate survival time, survival status, and expression data of all genes in tumor samples from colon cancer patients in the TCGA database. The Cox proportional hazards model was used to assess the impact of each gene on the prognosis of tumor patients. Following this, an intersection was taken among the pathologically differentially expressed genes, pharmacologically differentially expressed genes, and genes with survival significance to obtain the key genes involved in the intervention of norcantharidin. Finally, a protein–protein interaction analysis and KEGG enrichment analysis were conducted on the key genes to identify the core genes affected by norcantharidin intervention.

### 4.13. Animal Experiments

Female BALB/c nude mice (four weeks old, 13–15 g) were purchased from Yaokang Biotechnology Co., Ltd. (Beijing, China). All animal procedures complied with China’s Regulations on the Administration of Experimental Animals and were approved by the Animal Experiment Ethics Committee of Beijing University of Chinese Medicine (Ethics Approval Number: 2023112202-4301). To construct the transplanted tumor model in the nude mice, 1 × 10^7^ cells were injected subcutaneously after a week of adaptive feeding, and the weight of the nude mice was measured every two days. The model was successfully constructed when the tumor volume reached 100 mm^3^. The animals were divided into four groups (*n* = 6): saline, oxaliplatin, NCTD-5 mg/kg, and NCTD-10 mg/kg. All group treatments were administered via intraperitoneal injection every 2 days for a total of 14 days. In the detection of the targeting ability of liposomes, two groups (*n* = 3) were injected with 2 mg of IR-1061-NCTD liposome and IR-1061-FA-NCTD liposome through the tail vein, respectively. After 4, 8, 12, and 24 h, imaging was performed using the Animal-Living Imager. To investigate the effect of FA-NCTD liposome on the subcutaneous transplantation of colon cancer in nude mice, all animals were divided into six groups (*n* = 6). The animals were treated with saline, oxaliplatin, NCTD, NCTD liposome, and FA-NCTD liposome. All group treatments were administered via tail intravenous injection every 2 days for a total of 14 days.

### 4.14. Real-Time Quantitative PCR (RT–qPCR) Analysis

According to the manufacturer’s instructions, total RNAs were prepared from cells using a Total RNA extraction kit from Tiangen Biotech (Beijing) Co., Ltd. (Beijing, China). Reverse transcription was performed on equal amounts of total RNA (1 μg) using the ReverTra Ace qPCR RT Kit. RT-qPCR was performed using SYBR Green Real-time PCR Master Mix (Toyobo, Osaka, Japan). GAPDH was used as a reference gene. Primers from Guangzhou RiboBio Co., Ltd. (Guangzhou, China) sequences were used, as presented in Table 1.

### 4.15. Western Blot

Cells were placed on ice and lysed in RIPA buffer supplemented with protease inhibitors and phosphatase inhibitors. The supernatant was collected via centrifugation at 12,000 rpm at 4 °C for 20 min, and the protein concentrations were measured using the BCA Detection Kit. Protein samples were separated using SDS-PAGE gel and transferred onto an NC membrane. The NC membranes were blocked with 5% skim milk for 2 h. Then, the primary antibody of the target protein was added and incubated at 4 °C overnight. Next, the membranes were incubated with the secondary antibody at room temperature for 1.5 h. An imaging system was utilized to visualize the protein bands, and Image J 1.8 software was used to qualify the protein expression. The following antibodies were used: anti-Cleaved Caspase 3 (Proteintech, Wuhan, China), anti-TRAF5 (Proteintech, Wuhan, China), anti-Caspase 3 (Proteintech, Wuhan, China), anti-c-Myc (ABclonal, Wuhan, China), anti-NF-κB p65 (ABclonal, Wuhan, China), anti-Phospho-NF-κB p65 (ABclonal, Wuhan, China), anti-IκBα (ABclonal, Wuhan, China), anti-Phospho-IκBα (ABclonal, Wuhan, China), anti-β-actin (Bioss, Beijing, China), and anti-GAPDH (Proteintech, Wuhan, China).

### 4.16. Hematoxylin–Eosin Staining

The tissues were immersed in 4% paraformaldehyde overnight. Subsequently, the tissues were embedded in paraffin and sliced into 5 μm slices. For general histology, the sections were stained with hematoxylin and eosin (Solarbio, Beijing, China). Microscopy (Nikon, Tokyo, Japan) was used to detect sections, and images were captured in random fields.

### 4.17. Immunohistochemistry

Immunohistochemistry (IHC) was performed to detect specific antigens in paraffin-embedded tissue sections. The tissue samples were fixed in 10% neutral buffered formalin for 24 h and subsequently embedded in paraffin. Sections of 4–5 µm thickness were cut using a microtome and transferred to glass slides. For antigen retrieval, the slides were immersed in citrate buffer. Following cooling, sections were rinsed in phosphate-buffered saline (PBS) and treated with 3% hydrogen peroxide to block endogenous peroxidase activity. Non-specific binding was minimized by incubating the sections with 3%BSA at room temperature for 30 min. The primary antibody was applied and incubated overnight at 4 °C. Then, the secondary antibody was applied for 50 min at room temperature. Finally, 3,3′-diaminobenzidine (DAB), as a chromogenic substrate, was applied for microscopic examination.

### 4.18. Norcantharidin Liposomes Targeting Folate Receptors

Cholesterol, DSPE-PEG2000, and norcantharidin were weighed out and dissolved in 10 mL of a dichloromethane–methanol mixture (5:1 ratio) followed by sonication until the solution became clear. A lipid thin film was formed through rotary evaporation, and then the film was dried using nitrogen gas. The membrane was washed with PBS and dispersed through sonication. The dispersion was sequentially filtered through 0.45 μm and 0.22 μm membranes, and ultrafiltration centrifugation was used to remove free components. The liposomes were redissolved in pure water and the filtration and centrifugation steps were repeated. Afterward, the liposomes were lyophilized and stored. The liposomes were later dissolved in physiological saline for further use.

### 4.19. Statistics

Statistical data were analyzed using SPSS 22.0 or GraphPad Prism 9.0. All data were represented using mean ± standard deviation ($\bar{x}$ ± SD). An unpaired Student’s *t*-test was used to identify the differences between the two groups. One-way analysis of variance (ANOVA) was used to identify the differences among multiple groups. *p* < 0.05 was defined as statistically significant.

## 5. Conclusions

In summary, NCTD effectively inhibits the malignant proliferation of colon cancer cells by regulating the TRAF5/NF-κB signaling pathway and inducing programmed apoptosis, offering a novel treatment strategy. Additionally, FA-NCTD liposomes were prepared, providing a basis for the precise targeted delivery and release of NCTD.

## Figures and Tables

**Figure 1 ijms-26-01450-f001:**
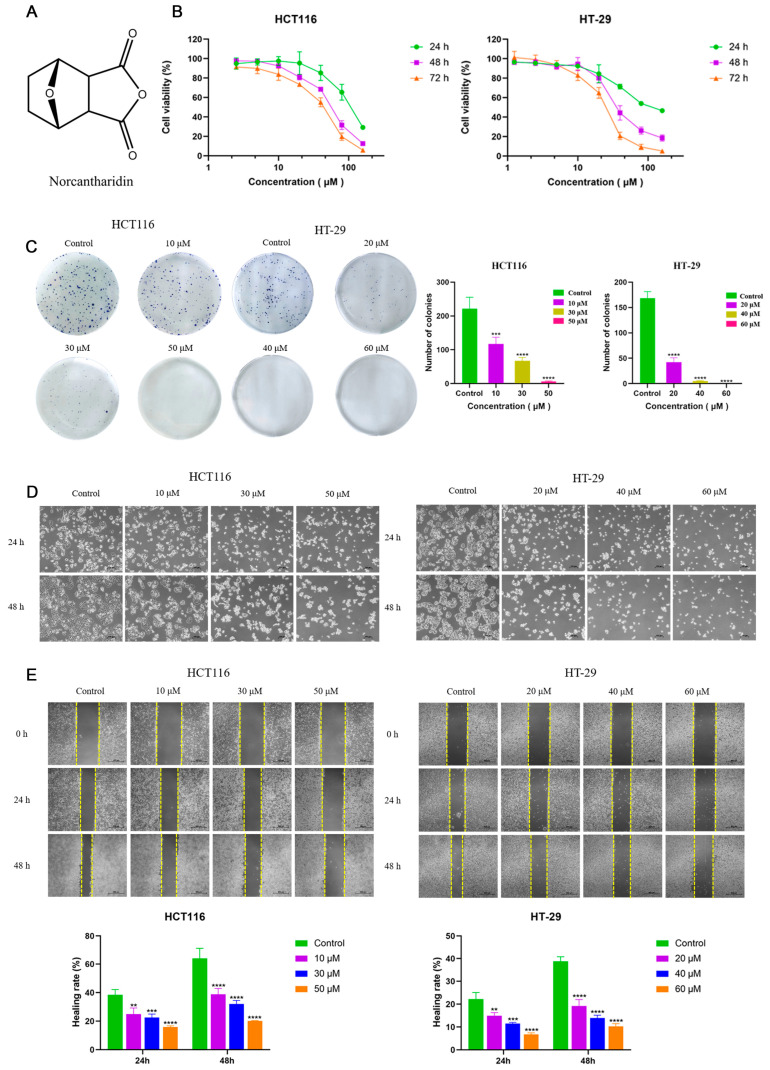
NCTD inhibited cell proliferation and migration in CRC cells. (**A**) The structure of norcantharidin. (**B**) The cell viability of HCT116 and HT-29 at 24 h, 48 h, and 72 h after NCTD treatment. (**C**) The colony-forming results for HCT116 and HT-29 exposed to NCTD. (**D**) The morphological changes in HCT116 and HT-29 cells exposed to NCTD for 24 h and 48 h (100 μm). (**E**) The effect of NCTD on wound healing was investigated in HCT116 and HT-29 cells (500 μm). At least three repetitions of each experiment were conducted. ** *p* < 0.01, *** *p* < 0.001, and **** *p* < 0.0001.

**Figure 2 ijms-26-01450-f002:**
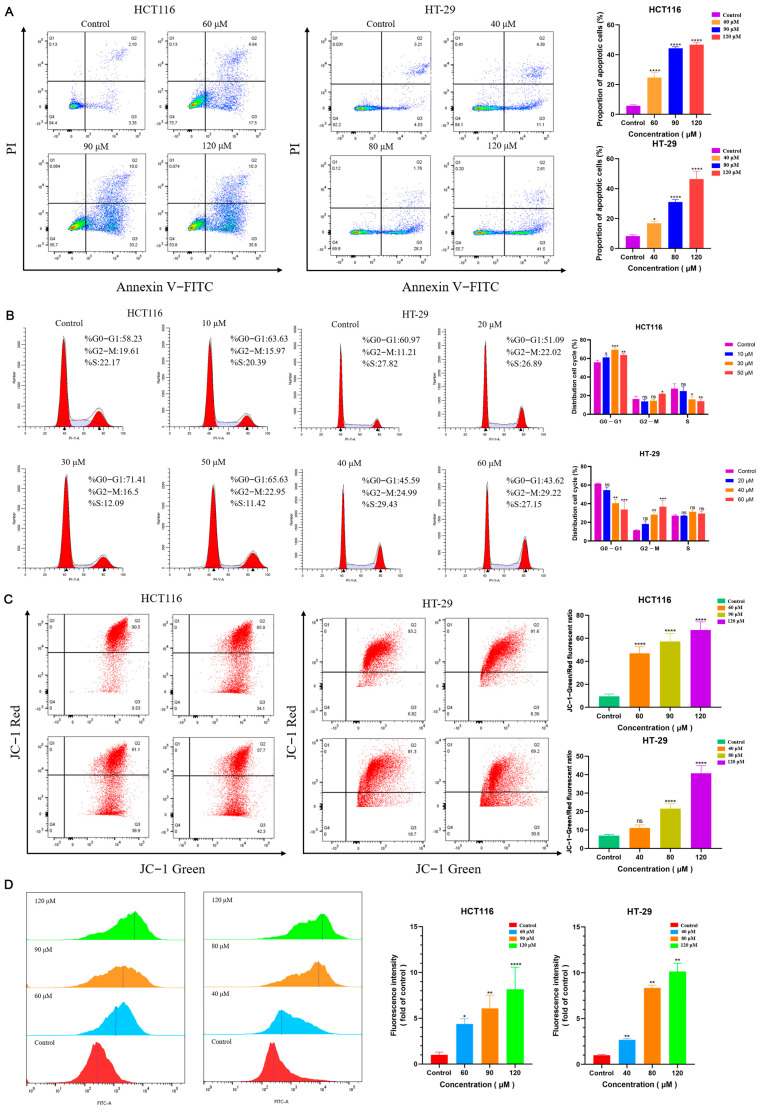
NCTD induced apoptosis and cell cycle arrest. (**A**) Apoptosis of HCT116 and HT-29 cells after NCTD treatment. (**B**) Cell cycle distribution of HCT116 and HT-29 cells after NCTD treatment. (**C**,**D**) Effect of NCTD on mitochondrial membrane potential and reactive oxygen species (ROS). At least three repetitions of each experiment were conducted. * *p* < 0.05, ** *p* < 0.01, *** *p* < 0.001, and **** *p* < 0.0001 and ns means not significant.

**Figure 3 ijms-26-01450-f003:**
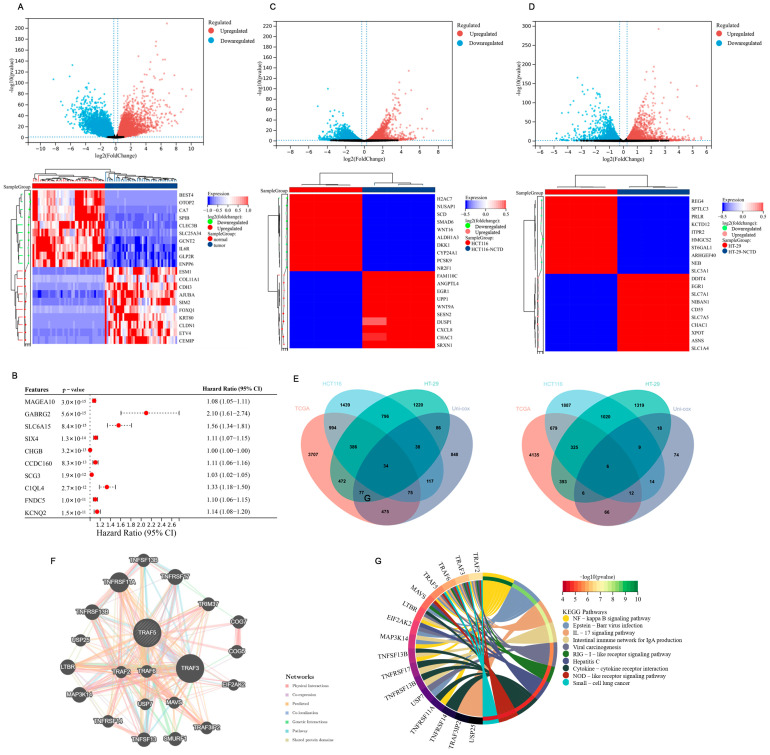
Bioinformatics analysis of NCTD intervention in HCT116 and HT-29 cells. (**A**) Upregulated and downregulated genes in tumor tissues compared to adjacent non-cancerous samples from TCGA database. (**B**) Cox proportional hazards model was employed to assess the impact of each gene on the prognosis of cancer patients. (**C**) Differential genes after NCTD treatment in HCT116. (**D**) Differential genes after NCTD treatment in HT-29. (**E**) Intersection of pathologically differential genes, pharmacologically differential genes, and genes with survival significance. (**F**,**G**) Protein–protein interaction analysis and Kyoto Encyclopedia of Genes and Genomes (KEGG) enrichment analysis of these key genes.

**Figure 4 ijms-26-01450-f004:**
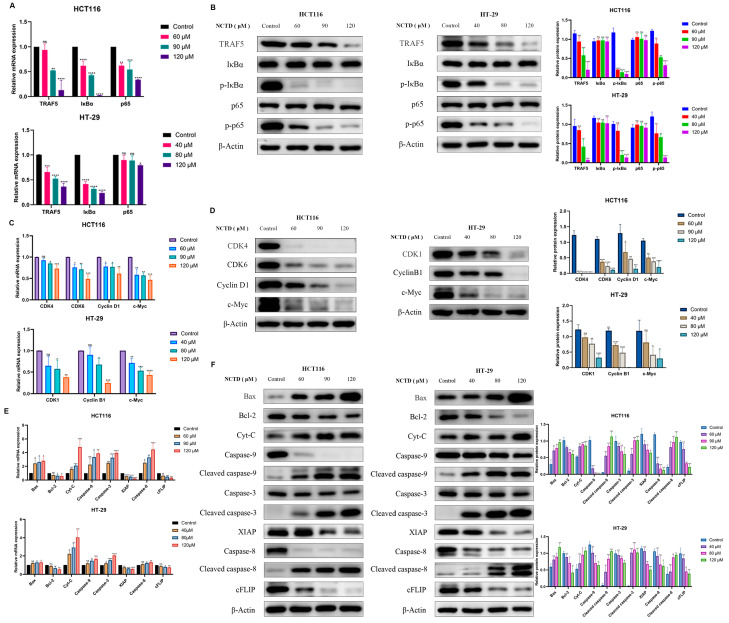
NCTD inhibits colon cancer by regulating the TRAF5/NF-κB signaling pathway. (**A**,**B**) The effect of NCTD on the TRAF5/NF-κB signaling pathway in HCT 116 and HT-29 cells. The mRNA and protein expression related to cell proliferation and apoptosis in NCTD-treated HCT116 and HT-29 cells (**C**–**F**). At least three repetitions of each experiment were conducted. * *p* < 0.05, ** *p* < 0.01, *** *p* < 0.001, **** *p* < 0.0001 and ns means not significant.

**Figure 5 ijms-26-01450-f005:**
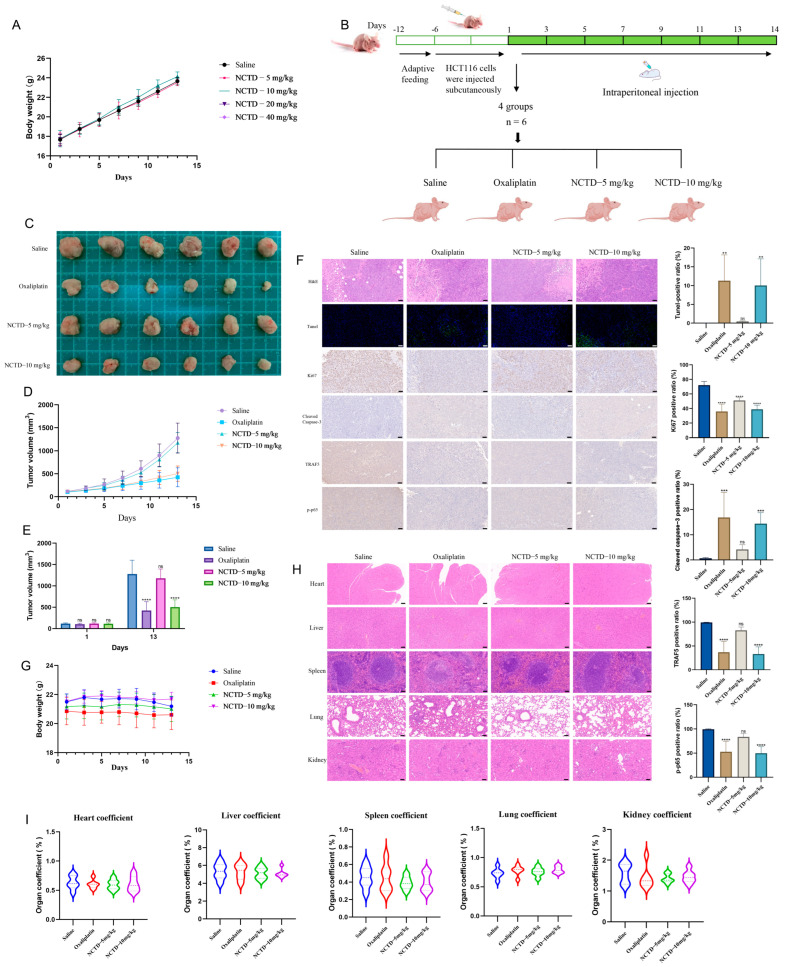
NCTD suppressed the progression of colon cancer transplanted in nude mice. (**A**) The effect of NCTD toxicity in nude mice (*n* = 3). (**B**) The administration of time, dosage, and method. (**C**–**E**) The effect of saline, oxaliplatin, NCTD-5 mg/kg, and NCTD-10 mg/kg on tumors (*n* = 6). (**F**) The Tunel-, Ki67-, cleaved caspase-3-, TRAF5-, and *p*-p65-positive cell number and ratio in saline, oxaliplatin, NCTD-5 mg/kg, and NCTD-10 mg/kg groups (50 μm, *n* = 6). (**G**) The changes in body weight in mice treated with saline, oxaliplatin, NCTD-5 mg/kg, and NCTD-10 mg/kg (*n* = 6). (**H**,**I**) The hematoxylin–eosin staining and organ coefficient in heart, liver, spleen, lung, and kidney (*n* = 6). ** *p* < 0.01, *** *p* < 0.001, and **** *p* < 0.0001 and ns means not significant.

**Figure 6 ijms-26-01450-f006:**
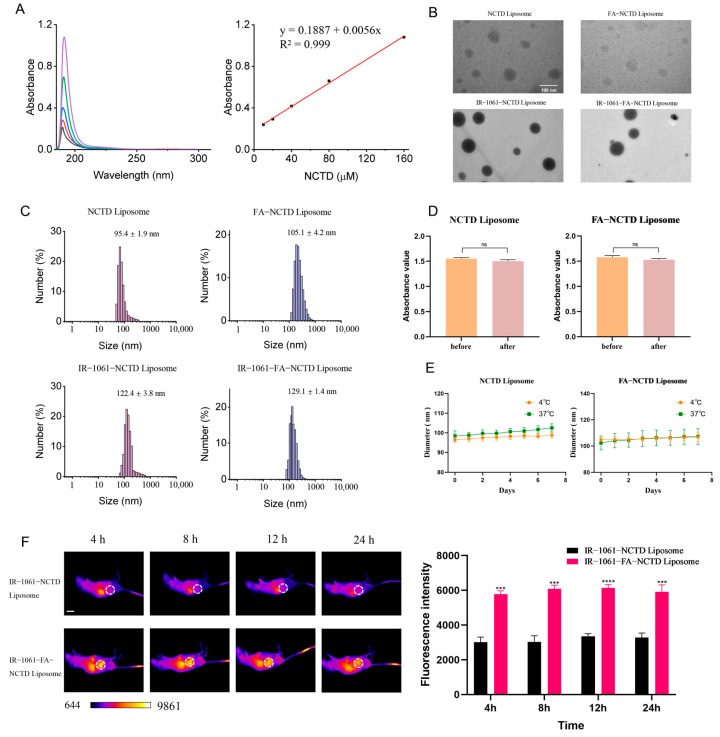
The preparation and quality evaluation of NCTD liposomes targeting folate receptors. (**A**) The maximum absorption wavelength and linearity of NCTD solution. (**B**) TEM images of NCTD, FA-NCTD, IR-1061-NCTD, and IR-1061-FA-NCTD liposomes (100 nm). (**C**) The particle size and Zeta potential of the liposomes were measured using a dynamic light scatterer (*n* = 3). (**D**) The absorbance values of NCTD and FA-NCTD liposomes before and after centrifugation (*n* = 3). (**E**) Changes in NCTD and FA-NCTD liposome diameter over 7 days at 4 °C and 37 °C (*n* = 3). (**F**) Target-finding ability of IR-1061-NCTD and IR-1061-FA-NCTD liposomes (1 cm, *n* = 3). *** *p* < 0.001, and **** *p* < 0.0001 and ns means not significant.

**Figure 7 ijms-26-01450-f007:**
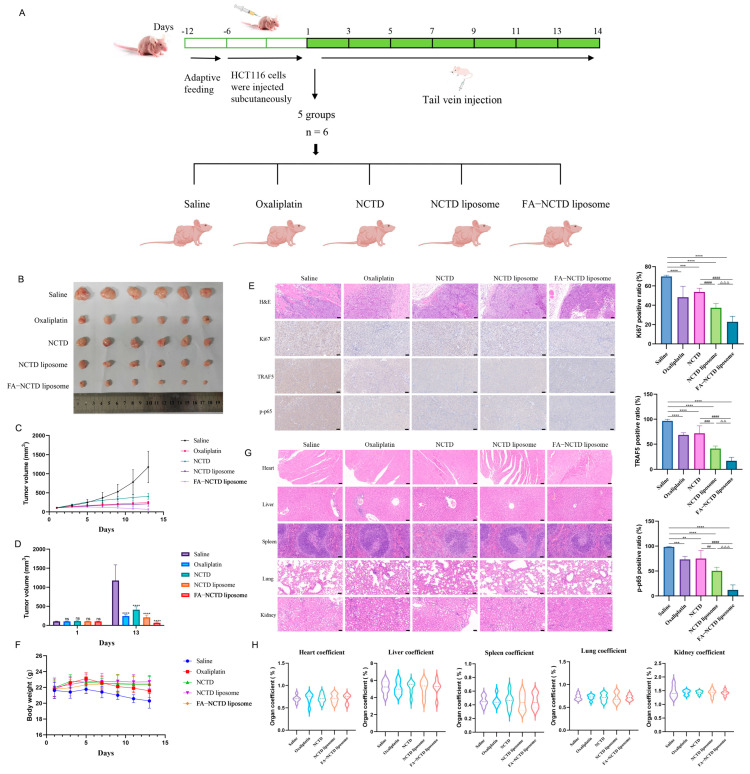
FA-NCTD liposomes suppressed the growth of colon cancer transplanted in nude mice. (**A**) Administration of time, dosage, method, and groups. (**B**–**D**) Effect of saline, oxaliplatin, NCTD, NCTD liposomes, and FA-NCTD liposomes on tumor (*n* = 6). (**E**) Ki67-, TRAF5-, and *p*-p65-positive cell number and ratio in saline, oxaliplatin, NCTD, NCTD liposome, and FA-NCTD liposome groups (50 μm, *n* = 6). (**F**) Changes in body weight of mice treated with saline, oxaliplatin, NCTD, NCTD liposomes, and FA-NCTD liposomes (50 μm, *n* = 6). (**G**,**H**) Hematoxylin–eosin staining and organ coefficient in heart, liver, spleen, lung, and kidney (*n* = 6). ** *p* < 0.01, *** *p* < 0.001, and **** *p* < 0.0001 compared with saline group. ^##^ *p* < 0.01, ^###^ *p* < 0.001, and ^####^ *p* < 0.0001 compared with NCTD group. ^△△^ *p* < 0.01 and ^△△△^ *p* < 0.001 compared with NCTD liposome group.

**Table 1 ijms-26-01450-t001:** Primer sequences used in real-time PCR.

Primers	Primer Sequence (5′-3′)
TRAF5	Forward CCGAGCCCCACAATGGCTTA Reverse CCGCTCCACAAACTGGTACT
Caspase-3	Forward CGTGTCATAAAATACCAGTGGA Reverse AAATTCTGTTGCCACCTTTCG
p65	Forward CCTGTCCTTTCTCATCCCATCTTTG Reverse GCTGCCAGAGTTTCGGTTCAC
IκBα	Forward GAGACTTTCGAGGAAATACCCC Reverse GTAGCCATGGATAGAGGCTAAG
CDK1	Forward TGCCGCTCTCCACCATCCG Reverse GCACACATCAAACAACCTGACCAC
CDK4	Forward TGAAATTGGTGTCGGTGCCTATGG Reverse CTCCTCCACCTCCTCCTCCATTG
CDK6	Forward TGCCGCTCTCCACCATCCG Reverse GCACACATCAAACAACCTGACCAC
Cyclin B1	Forward GCCAGTGCCAGAGCCAGAAC Reverse CATTGGGCTTGGAGAGGCAGTATC
Cyclin D1	Forward GCCCTCGGTGTCCTACTTCAAATG Reverse TCCTCCTCGCACTTCTGTTCCTC
c-Myc	Forward CCTGGTGCTCCATGAGGAGAC Reverse CAGACTCTGACCTTTTGCCAGG
GAPDH	Forward TGGAGTCCACTGGCGTCTTCAC Reverse TTGCTGATGATCTTGAGGCTGTTGTC

## Data Availability

Data are available upon request to the corresponding authors.

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
