# Peer review of "Exploring the Anti-Colorectal Cancer Mechanism of Norcantharidin Through TRAF5/NF-κB Pathway Regulation and Folate-Targeted Liposomal Delivery"

_ijms, 2025, doi:10.3390/ijms26041450_

Round 1
Reviewer 1 Report
Comments and Suggestions for Authors
The authors present an in vitro and in vivo study of norcantharidin (NCTD), a less toxic derivative of cantharidin. In addition, the authors present a liposomal formulation targeting the folate ligand (FA-NCTD) and characterize its properties, especially compared with oxaliplatin.
The work and the results are definitely of interest, but could be further improved with some modifications:
- High resolution figures (e.g. figure 7F shows visible artifacts) must be provided
- The RNA sequencing could give precious insight on the mechanism of action. However, it is not exactly clear how the cell lines were exposed to NCTD (concentration, duration) prior to sequencing. This should be clarified in the methods.
- The authors restrict their analysis of the transcriptome to genes that are prognostic or differentially expressed (DE) between tumor and normal, reducing the number of genes from several thousands to 40. This should be clearly justified. Why would the effects of NCTD only matter or be detectable on genes that are prognostic? Also, does the RNAseq show the same changes that are seen by RT-PCR?
- At a minimum, gene set enrichment analysis should be performed between the treated and untreated cell lines without reducing the DE genes and provided as a supplementary figure.
- The organ coefficients are good additions to the preliminary toxicological analysis. It may be useful to do formal statistical comparisons between the compounds (for example in figure 7H)
- The mouse experiments (figures 5B, 7A) are only based on HCT116, which is an MSI cell line that is not representative of all colorectal cancers. At a minimum, the HT-29 should also be used. Ideally, a more diverse panel of cell lines should be used when a new compound is studied.
Overall, this a convincing study that could benefit from some additional work to enhance the confidence in the results, especially as it could lead to clinical developments down the road.
Author Response
- High resolution figures (e.g. figure 7F shows visible artifacts) must be provided
Response: Thank you for your suggestion. We have been corrected to enhance the quality of the presentation. Please check.
- The RNA sequencing could give precious insight on the mechanism of action. However, it is not exactly clear how the cell lines were exposed to NCTD (concentration, duration) prior to sequencing. This should be clarified in the methods.
Response: Thanks for your suggestion. HCT116 and HT-29 cells were treated with 60 μM and 40 μM of norcantharidin for a duration of 48 hours, respectively. We added the description in methods (Line 461-462, Page 18). Thanks for your suggestion again.
- The authors restrict their analysis of the transcriptome to genes that are prognostic or differentially expressed (DE) between tumor and normal, reducing the number of genes from several thousands to 40. This should be clearly justified. Why would the effects of NCTD only matter or be detectable on genes that are prognostic? Also, does the RNAseq show the same changes that are seen by RT-PCR?
Response: Thank you for your thorough review and valuable feedback on our research. In response to your questions, we would like to provide detailed explanations and clarifications as follows:
Firstly, in this study, faced with the vast amount of differential gene data obtained through high-throughput sequencing, we adopted a rigorous screening strategy to ensure that the final selected genes possessed both biological significance and practical clinical relevance. We analyzed the differential genes between tumor tissues and normal tissues, which may directly participate in the occurrence and development of tumors. Additionally, we analyzed the differential genes produced after drug intervention (norcantharidin), reflecting the direct impact of the drug on tumor cells. Finally, we identified genes with prognostic significance, which are directly related to the prognosis of tumor patients. By taking the intersection of these three categories of genes, we obtained a gene set that not only reflected the effect of drug intervention but was also closely related to the prognosis of patients. This process not only ensured that the selected genes had clear biological functions and clinical significance but also greatly reduced the complexity and cost of subsequent experimental validation. Ultimately, we identified 40 genes with prognostic significance after norcantharidin intervention, which are important candidates for further research and potential clinical applications. This screening strategy not only improved the pertinence and accuracy of the study but also provided a solid foundation for subsequent functional validation, mechanism exploration, and clinical application.
Secondly, prognostic genes play a crucial role in tumor research. They provide key information about the biological characteristics of patients' tumors and their response to treatment, helping doctors more accurately predict the prognosis of patients. This, in turn, guides the formulation of personalized treatment plans, optimizes treatment outcomes, and provides a scientific basis for the long-term management of tumor patients. In this study, we obtained a large number of genes through high-throughput sequencing, including both pathological and pharmacological genes, which contained both prognostic and non-prognostic genes. To improve the pertinence and accuracy of the study, we screened and studied the prognostic genes affected by norcantharidin intervention.
Thirdly, this study focuses on the impact of norcantharidin on TRAF5. High-throughput sequencing results showed that TRAF5 was highly expressed in tumors, and its expression level decreased after treatment with norcantharidin. RT-qPCR experiments further confirmed that norcantharidin could significantly reduce the expression level of TRAF5. Thanks for your suggestion again.
- At a minimum, gene set enrichment analysis should be performed between the treated and untreated cell lines without reducing the DE genes and provided as a supplementary figure.
Response: Thank you very much for your valuable advice and guidance. In response to your suggestion to supplement the Gene Set Enrichment Analysis (GSEA), we have carefully considered and conducted additional analyses, and have compiled the results into supplementary figures (Line 150-160 Page 6). Thank you once again for your suggestion.
- The organ coefficients are good additions to the preliminary toxicological analysis. It may be useful to do formal statistical comparisons between the compounds (for example in figure 7H)
Response: Thanks for your suggestion. We agree with you very much. However, there were no significant changes in organ coefficient in the heart, liver, spleen, lung and kidney (Figure 7H). Statistical analysis found no significant difference between the groups. Thank you for your suggestion again.
- The mouse experiments (figures 5B, 7A) are only based on HCT116, which is an MSI cell line that is not representative of all colorectal cancers. At a minimum, the HT-29 should also be used. Ideally, a more diverse panel of cell lines should be used when a new compound is studied.
Response: Thanks for your suggestion. We agree with you. HCT116 and HT-29 are two distinct human colorectal cancer cell lines. HCT116 is an MSI cell line while HT-29 is MSS. In previous study, it has been reported that norcantharidin inhibits the growth of HT-29 transplanted tumors (PMID: 26187792). Therefore, we chose HCT116 cells in animal experiment. Thanks for your suggestion again.
Reviewer 2 Report
Comments and Suggestions for Authors
This manuscript found that NCTD inhibits multiple activities of cancer cells through ex vivo and in vivo experiments, whole transcriptome sequencing and so on. A more systematic investigation was carried out, which has certain innovation and research value. To improve the manuscript, I recommend the authors address the following points:
1. NCTD regulates the TRAF5/NF-κB signaling pathway to exert anti-colorectal cancer effects. Other related molecular mechanisms upstream and downstream of this pathway are under-explored in the manuscript.
2. It is suggested to add other colon cancer cell lines with different characteristics to enhance the generalizability of the findings.
3. Only the fluorescence intensity of fluorescently labeled liposomes at the tumor site was verified, which is a single method. It is suggested to add other experimental methods to further validate its targeting.
4. It is suggested to add the analysis of potential challenges and opportunities for future clinical translation of NCTD and its liposomes in the discussion section.
5. Comparative data with commonly used drugs should be added to highlight the advantages or differences of NCTD and its liposomes in terms of efficacy and toxicity.
6. The labeling of some bar charts in Figure 1 and Figure 2 is not clear enough.
7. It is recommended to optimize the quality of Figure 3 to ensure that the data in the charts are clear and readable.
Author Response
- NCTD regulates the TRAF5/NF-κB signaling pathway to exert anti-colorectal cancer effects. Other related molecular mechanisms upstream and downstream of this pathway are under-explored in the manuscript.
Response: Thank you for your valuable comments and corrections on our work. We take seriously the issue you raised regarding the insufficient exploration of the upstream and downstream molecular mechanisms in our study on the anti-colorectal cancer effects of NCTD via regulating the TRAF5/NF-κB signaling pathway. Indeed, in this research, we primarily focused on how NCTD inhibits the growth and metastasis of colorectal cancer by modulating the TRAF5/NF-κB signaling pathway. However, we are also aware that a complex signaling pathway often involves multiple upstream and downstream molecules and intricate interaction networks. To gain a more comprehensive understanding of the mechanism of action of NCTD, it is essential to further explore these upstream and downstream molecular mechanisms. In future studies, we plan to adopt various experimental approaches, such as gene knockout, overexpression, RNA interference, and protein interaction analysis, to delve deeper into the key molecules and their interactions upstream and downstream of the TRAF5/NF-κB signaling pathway. Additionally, we will utilize high-throughput sequencing and bioinformatics analysis to reveal changes in the proteome of cells before and after NCTD treatment, thereby discovering more molecular mechanisms related to the action of NCTD. We sincerely hope that these follow-up studies will further enrich and refine our understanding of the anti-colorectal cancer mechanism of action of NCTD. At the same time, we also look forward to continuing to receive your attention and guidance in our future work. Thank you once again for your valuable comments.
- It is suggested to add other colon cancer cell lines with different characteristics to enhance the generalizability of the findings.
Response: Thanks for your suggestion. At present, the commonly used colon cancer cell lines mainly include HCT116, HT-29, SW480, SW620, Caco-2 and DLD-1. Different cell lines possess unique genetic backgrounds, phenotypic traits, and biological characteristics, each providing specific advantages in research applications. HCT116 and HT-29 cell lines are well-suited for studies on drug mechanisms. SW480 and SW620 are well-suited for metastatic colon cancer. Caco-2 often used in drug absorption and metabolism studies, it can form glandlike structures and simulate the intestinal epithelial barrier. DLD-1 is used to study the genetic background of colon cancer and its response to drugs. Therefore, HCT116, HT-29 were selected as the cell lines for this study. Thanks for your suggestion again.
- Only the fluorescence intensity of fluorescently labeled liposomes at the tumor site was verified, which is a single method. It is suggested to add other experimental methods to further validate its targeting.
Response: Thank you very much for your thorough review and valuable feedback on our work. We should conduct a more comprehensive validation of the targeting efficiency of the liposomes. For instance, collecting cells from both tumor tissue and surrounding non-tumor tissue and analyzing the content of fluorescently labeled liposomes in these cells through flow cytometry is a promising approach. However, due to the fact that the fluorescent molecule we used is IR1061 in the NIR-II (900-1700 nm), our current instruments are unable to meet the experimental requirements, and thus we did not proceed with further validation. Additionally, our pharmacodynamics studies have shown that the therapeutic effect of the folate receptor-targeted liposome group is more significant compared to the non-targeted liposome group. These results all indicate that targeted liposomes can accumulate at the tumor site, achieving a more desirable therapeutic outcome.
- It is suggested to add the analysis of potential challenges and opportunities for future clinical translation of NCTD and its liposomes in the discussion section.
Response: Thanks for your suggestion. We added the analysis of potential challenges and opportunities for future clinical translation of NCTD and its liposomes in the discussion section (Line 395-402 Page 16). Please check.
- Comparative data with commonly used drugs should be added to highlight the advantages or differences of NCTD and its liposomes in terms of efficacy and toxicity.
Response: Thanks for your suggestion. The most common chemotherapy regimens for colorectal cancer are oxaliplatin, fluorouracil and irinotecan complemented by radiotherapy, immunotherapy, and molecular-targeted strategies (Line 299-303 Page 15). However, drug resistance and side effects are the main problems in clinical application. Norcantharidin (NCTD) demonstrates reduced toxicity while retaining similar antitumor efficacy. Many studies have revealed that NCTD exerts its antineoplastic effects by modulating diverse signaling pathways and inhibiting the progression of multiple malignancies. This study investigated the mechanism of norcantharidin against colorectal cancer. In addition, this study aims to develop targeted liposomes that incorporate tumor cell-specific ligands, enhancing the accumulation of drugs at the tumor site for more effective anti-cancer treatment. Furthermore, targeted liposomes containing NCTD may exhibit synergistic effects when used in combination with other anticancer drugs, thereby enhancing overall efficacy. Norcantharidin has fewer side effects than drugs such as oxaliplatin, fluor-ouracil and may improve patient tolerance to the drug. However, more extensive clinical studies are still needed to verify its efficacy and safety in the treatment of colorectal cancer (Line 399-402 Page 16). Thanks for your suggestion again.
- The labeling of some bar charts in Figure 1 and Figure 2 is not clear enough.
Response: Thanks for your suggestion. We have revised Figure 1 and 2. Please check. Thank you for your suggestions again.
- It is recommended to optimize the quality of Figure 3 to ensure that the data in the charts are clear and readable.
Response: Thanks for your suggestion. We have optimized the quality of Figure 3 and examined all figures. Please check. Thank you for your suggestions again.
Reviewer 3 Report
Comments and Suggestions for Authors Developing liposomal targeted delivery systems involves attaching targeting molecules, such as antibodies or ligands, to the surface of liposomes. This modification allows the liposomes to actively recognize and bind to tumor cells, facilitating efficient drug delivery. Due to the overexpression of folate receptors on the surfaces of tumor cells, these receptors have become significant targets for cancer therapy. In this study, the authors utilized PEG modification technology and a folate-targeted strategy to design and synthesize actively targeted liposomes loaded with norcantharidin (NCTD). The folate ligand on the liposome surface specifically targets and binds to folate receptors on colon cancer cells, ensuring precise drug delivery and release. This approach aims to enhance the antitumor efficacy of NCTD while minimizing its potential toxic effects on non-target tissues.The strength of the manuscript: the application of modern technology of targeted drug delivery by liposomes with ligands that is specific for a certain tissue. A large number of figures that facilitate the monitoring of results and discussions.
Weakness of the manuscript: The figures are too complex, so the details are difficult to see. Suggested minor corrections: 1. In the introductory part, when the authors discuss the structure of Cantharidin, they state that by removing the methyl group from positions 1 and 2, Norcantharidin is obtained, it is necessary to present the molecular structure of Cantharidin, and the reaction to obtain Norcantharidin, that is the only way it makes sense to talk about the positions in to the structure of Cantharidin. 2. Give the general scheme of the liposome, indicate where the ligand for tissue recognition is located and indicate the position of the active component, ie. medicine. 3. Some parts of large pictures should be enlarged, since the details cannot be seen, for example Figure 2 C,D; Figure 3, Figure 4, Figure 5. 4. Emphasize what is new in the conclusion.
Author Response
- In the introductory part, when the authors discuss the structure of Cantharidin, they state that by removing the methyl group from positions 1 and 2, Norcantharidin is obtained, it is necessary to present the molecular structure of Cantharidin, and the reaction to obtain Norcantharidin, that is the only way it makes sense to talk about the positions in to the structure of Cantharidin.
Response: Thank you for your suggestion. Cantharidin and Norcantharidin have different structural characteristics (see the following figure). Norcantharidin is usually obtained by the following reaction. In Figure 1, we added the structural formula of norcantharidin. Norcantharidin is not produced by the removal of the methyl groups at positions 1 and 2 of cantharidin. We also revised the introduction. Thank you for your suggestion again.
- Give the general scheme of the liposome, indicate where the ligand for tissue recognition is located and indicate the position of the active component, ie. medicine.
Response: Thanks for your suggestion. Liposomes are spherical vesicles formed by phospholipids, resembling cell membranes, and are self-assembled from amphiphilic molecules with hydrophilic heads and hydrophobic tails. Ligands are covalently attached to the outer layer of the phospholipid bilayer, ensuring effective binding to receptors on the cell membrane when liposomes come into contact with target cells. The active ingredient is typically encapsulated within the hydrophobic interior of the liposome. This encapsulation not only protects the drug from external environmental interference but also ensures that it retains its biological activity when it reaches the target cell. Upon fusion of the liposome with the target cell, the drug is released from the liposome and enters the cell to exert its therapeutic effect.
- Some parts of large pictures should be enlarged, since the details cannot be seen, for example Figure 2 C,D; Figure 3, Figure 4, Figure 5.
Response: Thanks for your suggestion. We optimized the quality of Figures, to enhance the visibility of the details. These adjustments will ensure that the important features are more readily observable. Thank you for helping us improve the quality of our manuscript. Thanks for your suggestion again.
- Emphasize what is new in the conclusion.
Response: Thanks for your suggestion. We have revised the discussion section. The discussion section has been expanded to include an analysis of the potential challenges and opportunities for the future clinical translation of NCTD and its liposomal formulations (Line 395-402 Page 16). Thanks for your suggestion again.
Round 2
Reviewer 2 Report
Comments and Suggestions for Authors
The quality of the paper has been further improved, and the proposed review comments were well responded to and revised. It is recommended for acceptance.